# Photoreduction of Copper Ions Using Silica–Surfactant Hybrid and Titanium (IV) Oxide under Sulfuric Acid Conditions

**DOI:** 10.3390/ma15155132

**Published:** 2022-07-24

**Authors:** Shingo Machida, Reo Kato, Kaishi Hasegawa, Takahiro Gotoh, Ken-ichi Katsumata, Atsuo Yasumori

**Affiliations:** 1Department of Material Science and Technology, Faculty of Advanced Engineering, Tokyo University of Science, 6-3-1 Niijuku, Katsushika-ku, Tokyo 125-8585, Japan; 8218b03@alumni.tus.ac.jp (R.K.); 8221543@ed.tus.ac.jp (K.H.); k.katsumata@rs.tus.ac.jp (K.-i.K.); yasumori@rs.tus.ac.jp (A.Y.); 2Material Characterization Central Laboratory, School of Science and Engineering, Waseda University, 3-4-1 Okubo, Shinjuku-ku, Tokyo 169-8555, Japan; tgotoh@waseda.jp

**Keywords:** mesoporous silica, silica–surfactant hybrid, heavy metal ion, photoreduction, titanium oxide

## Abstract

Photoreduction of Cu^2+^ ions to Cu metal by titanium(IV) oxide (TiO_2_) was conducted in the presence of a silica–surfactant hybrid under sulfuric acid conditions. After irradiation, a dark-red color, reflections due to Cu metal in the X-ray diffraction pattern, and peaks due to Cu 2*p*_1/2_ and 2*p*_3/2_ in the X-ray photoelectron spectrum indicated the precipitation of Cu metal in the product. In addition, an increase in the Brunauer–Emmett–Teller specific surface area from 36 and 45 m^2^/g for the silica–surfactant and TiO_2_, respectively, to 591 m^2^/g for the product, and a decrease in the intensity of the C-H stretching band in the Fourier–transform infra-red spectra implied the removal of surfactant during the reaction. These characteristics were never observed when TiO_2_ was used solely. Therefore, this study indicated that the photoreduction of Cu^2+^ ions to Cu metal by TiO_2_ was facilitated under the sulfuric acid medium, where the surfactants extracted from silica–surfactant hybrids by protons in the acidic condition were successfully photo-oxidized by TiO_2_. Thus, this study presents a new application of the conversion of a silica–surfactant hybrid into mesoporous silicas.

## 1. Introduction

The discovery of mesoporous silicas prepared by a supramolecular templating approach in the 1990s [1,2] prompted studies of their fundamental properties and practical applications, leading to the generation of mesoporous silicas with varied mesostructures that originated from the mesostructures of silica–surfactant hybrids that form via self-assembly of surfactants such as alkyltrimethylammonium salts with concurrent silica precipitation [1,2,3,4,5,6,7]. Thus, porous structures form upon removal of the surfactants via calcination using an electric furnace or via acid extraction under mild conditions [1,2,3,4,5,6,7]. The former process can emit carbon dioxide gas, whereas the latter process proceeds under mild conditions but generates a waste solvent that contains surfactants. Therefore, we focused on adding this surfactant-containing waste to wastewater to promote the removal of pollutants for environmental purification applications. Notably, mesoporous silicas that release surfactants can be used as adsorbents to purify waste water [6,7].

Here, we report the photoreduction of Cu^2+^ ions using silica–surfactant hybrids and titanium (IV) oxide (TiO_2_) under sulfuric acid conditions (Figure 1). Specifically, photoreduction of heavy-metal ions was conducted with the concurrent oxidation of surfactants by combining TiO_2_ with silica–surfactant hybrids. TiO_2_ is an extensively studied photocatalyst [8,9,10] that promotes the photo-oxidation of organic compounds and the photoreduction of heavy-metal ions to metals [8,9,10,11]. The latter process can be used to produce visible-light-responsive photocatalysts and antibacterial materials [8,9,10,11], and the generated heavy metals can be photo-oxidized by TiO_2_ [12]. Notably, heavy-metal ions are present in sulfuric-acid-containing wastewater from mines [13,14]. Sulfate ions can be adsorbed onto the surface of TiO_2_ to decrease its photocatalytic activity [15]. Therefore, surfactants extracted from silica–surfactant hybrids by sulfuric acid are promising organic compounds that can be degraded by photo-oxidation by TiO_2_, promoting the photoreduction of heavy-metal ions by TiO_2_ in the presence of sulfuric acid. Thus, the photocatalytic activity of TiO_2_ has the potential to both degrade surfactants present as additional waste and remove heavy-metal ions as metal precipitates.

## 2. Material and Methods

Copper(II) sulfate (CuSO_4_) pentahydrate and methanol were obtained from Wako Pure Chemical. Tetraethoxysilane (TEOS) and hexadecyltrimethylammonium chloride (C16TAC) were obtained from TCI. A 28 wt% ammonia solution was obtained from Kanto Chemical (Tokyo, Japan). All the chemicals were reagent grade and used without further purification. 

In the present study, we conducted photoreduction of Cu^2+^ ions because their concentration is easily measured and estimated by UV–vis spectrophotometric analysis of the well-known blue-colored complex [Cu(NH_3_)_4_]^2+^ [16]. To easily determine the particle morphology, in accordance with a previous study [17], we prepared spherical silica–surfactant hybrid particles via homogeneous precipitation of both silica and C16TAC in a methanolic solution containing ammonia. After TEOS was added to the methanol/water mixture containing ammonia (the TEOS:C16TAC:H_2_O:methanol:ammonia molar ratio was 1:0.4:774:1501:72 in the initial dispersion), the solution was shaken for 3 s and aged at room temperature for 20 h. After the reaction, the resultant solid was centrifuged, washed with methanol, and then dried for 80 °C for 1 day. A field-emission scanning electron microscopy (FE-SEM) image of the product is shown in Figure 1. For comparison, the as-synthesized spherical particles (ASP) were calcined at 550 °C for 20 h, similar to the procedure used in a previous study [17] to prepare calcined spherical particles (CSP) as a nanoporous silica. The TiO_2_ was a standard photocatalyst (P25, Degussa, Düseldolf, Germany). After both P25 (20 mg) and ASP (80 mg) were dispersed in a 10 mmol/L CuSO_4_ solution (5 mL) adjusted to pH 4 by the addition of sulfuric acid, the dispersion was irradiated with a He–Xe lamp for 4 h though a quartz plate. The irradiation area was the same as the width of the reaction vessel, and the distance between the top of the dispersion (whose height from the bottom of the reaction vessel was 1 cm) and the lamp was 0.5 cm. During irradiation, the dispersion was briefly stirred for every hour. The dispersion was also continuously stirred for comparison purposes. After the reaction, the resultant solid was centrifuged at 5000 rpm for 10 min and then washed with distilled water, to obtain the product denoted herein as Cu–P25–ASP. The resultant supernatant showed an increase in pH from 4 to 5 and contained bubbles on its surface, a common feature of surfactant aqueous solutions. The obtained solid was dried under a reduced pressure for 1 day. For comparison, these procedures were also applied to separate starting solutions containing P25 (20 mg), ASP (80 mg), and C16TAC (20 mg) individually.

## 3. Results and Discussion

After irradiation, the P25–ASP mixed dispersion was dark-red (Figure 2), whereas the dispersions of P25 or ASP and the solution of C16TAC containing Cu^2+^ were not. Notably, when the P25-ASP mixed dispersion was continuously stirred, the red coloration was not observed. Because the powders used in the present study did not immediately settle in the reaction vessel, UV light likely did not reach the vessel bottom and a certain UV irradiation time was necessary to change the dispersion color. The X-ray diffraction (XRD) pattern for the Cu–P25–ASP shows reflections attributable to P25 and ASP as well as Cu metal; no reflections attributable to copper oxides are observed (Figure 3). Transmission electron microscopy (TEM) and scanning TEM (STEM) images of the Cu–P25–ASP show spherical particles (ASP) and ~30 nm particles of P25. The latter particles exhibit relatively dark and bright regions (Figure 4a,b). The light regions contain high concentrations of Cu, as evident in the corresponding energy-dispersive X-ray (EDX) mapping image (Figure 4d). In addition, the spherical particles and the ~30 nm particles of P25 contain Si and Ti, as revealed in the EDX mapping images, respectively (Figure 4e,f). The X-ray photoelectron spectrum of the Cu–P25–ASP (Figure 5) shows peaks at binding energies of 953.5, 943.0, 933.0, and 932.5 eV, which are attributed to Cu 2*p*_1/2_, Cu^2+^ satellite, Cu^2+^ 2*p*_3/2_, and Cu 2*p*_3/2_, respectively [18,19]. The ratios of the decrease in Cu^2+^ concentration relative to those in the starting solutions are reported in Table 1. The dispersion that contained both P25 and ASP resulted in the largest decrease, whereas the Cu^2+^ concentration hardly decreased for the P25 dispersion or the C16TAC solution. In contrast, the ASP dispersion resulted in a 40% decrease in Cu^2+^ concentration.

The intensities of the C–H stretching bands [20] at 2922 and 2872 cm^−1^ and the C–N stretching band [20] at 1460 cm^−1^ in the Fourier transform infrared (IR) spectrum of the solid mixture of P25 (20 mg) and ASP (80 mg) are substantially lower than those in the spectrum of the Cu–P25–ASP (Figure 6). Based on the CHN analyses, the C and N contents (2.9 and 0.13 mass%, respectively) in the Cu–P25–ASP are lower than those in the mixed solid before irradiation (18 and 0.95 mass%, respectively). Figure 7 shows N_2_ adsorption/desorption isotherms for the ASP, P25, Cu–P25–ASP, and CSP, recorded at −196 °C. Based on these isotherms, the specific surface areas of the ASP, P25, Cu–P25–ASP, and CSP were calculated to be 36, 45, 591, and 806 m^2^/g using the Brunauer–Emmett–Teller (BET) method [21]. These BET surfaces areas, the Barrett–Joyner–Halenda (BJH) [22] pore sizes, and the pore volumes estimated using the BJH method are listed in Table 2 for Cu–P25–ASP and CSP.

Collectively, the product appearance (Figure 2), XRD patterns (Figure 3), XPS spectrum (Figure 5), and electron microscopy and elemental analysis results (Figure 4) reveal that Cu^2+^ was photo-reduced, resulting in the precipitation of Cu metal in the P25–ASP mixed product after irradiation. Copper oxides are absent, consistent with the lower redox potential of the conduction band of TiO_2_ (−0.52 V) [8] relative to those for Cu^2+^/Cu^+^ (0.15 V), Cu^2+^/Cu (0.34 V), and for Cu^+^/Cu (0.52 V) [23].

Alkylammonium ions are well known to be degraded by ^•^OH and ^•^O_2_ radicals, which can be generated by reaction of H_2_O and O_2_ molecules with holes photogenerated in the valence band of TiO_2_ [24]. In the present study, the IR spectra, N_2_ adsorption/desorption isotherms (Figure 7), and elemental analyses indicate a decrease in the concentration of hexadecyltrimethylammonium ions (C16TMA^+^) in the ASP after the reaction. Because a decrease in the concentration of Cu^2+^ was observed when ASP alone was used for the present reaction (Table 1) and because the cation-exchange reactions of silanol groups on silicas are well known [25], exchange reactions of C16TMA^+^ with protons and/or Cu^2+^ ions in the CuSO_4_ acidic solution are highly likely, as indicated by an increase in the pH of the dispersion after the reaction (see Material and Methods). The presence of Cu^2+^ detected by XPS (Figure 5) thus results from Cu^2+^ ions adsorbed onto the ASP, which are not accessible on P25 surfaces (Figure 4) or on the white regions observed after irradiation (Figure 2); the latter situation can be improved by modifying the experimental conditions. The dispersion never became dark-red as shown in Figure 2 when the present reaction was conducted with P25 alone; it remained white. In addition, the Cu^2+^ decrease ratio for the Cu–P25–ASP is higher than those for the P25 and ASP. Notably, the solution after irradiation was still under sulfuric acid conditions (refer to the experimental conditions). Therefore, photo-oxidation of C16TMA^+^ extracted from ASP is evident in the present reaction, and this photo-oxidation successfully promoted the photoreduction of Cu^2+^ under sulfuric acid conditions. 

After the dispersion containing P25 and ASP was irradiated for 4 h, slight bubbling was observed on the surface of the dispersed particles, suggesting the presence of undegraded C16TMA^+^. In addition, C16TMA^+^ and/or C16TAC was incompletely extracted from the ASP according to the residual alkyl chains in Cu–P25–ASP, as revealed by the IR spectra and CHN analyses; this interpretation is further supported by the lower porosity of Cu–P25–ASP compared with that of CSP, as determined from the N_2_ adsorption/desorption isotherms (Figure 7). Notably, silica–surfactant hybrid spheres have been prepared using alkyltrimethylammonium salts with shorter side-chain lengths [4]. In addition, the synthesis of silica–surfactant hybrids has been used to coat silica–surfactant layers onto various inorganic surfaces to form inorganic solid/silica–surfactant core–shell particles [5,17]. Increasing the amounts of TiO_2_ and TiO_2_-based compounds [26,27] and/or other materials [28] added to the present reaction is also feasible. Meanwhile, because the Cu^2+^ adsorption capability of P25 is low under sulfuric acid conditions (Table 1), the cation adsorption ability of TiO_2_ surfaces in the presence of protons, where acid sites on TiO_2_ surfaces could be strongly related to their adsorption ability, warrants further investigation [29,30,31,32]. However, the present product containing Cu metal might find applications where visible-light-responsive photocatalysts and antibacterial materials [7,8,9,10,11,12,13,14] have been used. We plan to conduct such studies in the future.

## 4. Conclusions

We demonstrated the photoreduction of Cu^2+^ ions to Cu metal by TiO_2_ under sulfuric acid conditions, where the surfactant molecules extracted from silica–surfactant hybrid spheres were successfully photo-oxidized by TiO_2_. Therefore, the present study might provide a new strategy for environmental purification, where the addition of another waste to wastewater efficiently promotes both degradation and removal of surfactants and heavy metals.

## Data Availability

Available on request from the corresponding author.

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
