# Peer review of "Photoreduction of Copper Ions Using Silica–Surfactant Hybrid and Titanium (IV) Oxide under Sulfuric Acid Conditions"

_materials, 2022, doi:10.3390/ma15155132_

Round 1
Reviewer 1 Report
The authors like to combine reduction of heavy metal ions and oxidation of surfactants both photocatalytically with TiO2 as the photocatalyst. This general approach is ok, but the whole story is not well presented. There are several lacks and I cannot recommend this manuscript for publication. This manuscript has not been prepared with the care and detail expected. Below are some comments that might help to improve the qualtity of the manuscript for future submission elsewhere.
Introduction:
[1] The reduction of heavy metals is clear, but why surfactants extracted from silica–surfactant hybrids as organic material and not surfactants in general? If only the surfactant should be degraded that comes from the synthesis (liquid phase) of porous silica, why where silica particles are used in the study?
[2] The target should be formulated more clear. "We like to reduce heavy metal ions and oxidize surfactants simultaneously by photocatalysis with TiO2 as the photocatalyst."
Materials:
[3] Most information are missing. Quality and origin of used chemicals.
[4] Why was the solution during irradiation not continuously stirred? Particles might settle and will not be irradiated between.
[5] The photocatalytic experiment must be explained in much more detail. The results depend strongly on the chosen conditons. Distance lamp-reactor, top- and side-irradiation, ...
[6] How was the concentration of cupper and surfactant measured? It is mentioned to complex Cu2+ with ammonia. With ammonia gas? Or ammonia solution? Was it measured with UV/vis spectrometry? Which machine, ...?
[7] Why not ICP for Cu2+?
[8] How was the concentration of the surfactant measured?
Results:
[9] Figure 3, JCPDS for P25 (one for the anatase and one for the rutile phase)
[10] Figure 4, When Cu-P25-ASP is a composite (adsorbed Cu2+), then the EDX mapping should be also done for Si and Ti.
[11] There is no Figure 6 as refered in the text. It is named Figure S2, obviously taken from any Supporting Information.
[12] All data from N2 adsorption (BET, pore size and volume) should be presented in a Table.
[13] In the text, elementary analysis and IR spectra are mentioned that show the removal of the surfactant. However, this manuscript does not contain the data from IR and elementary analysis.
[14] After the dispersion containing P25 and ASPs was irradiated for 4 h, slight bubbling was observed. And where does the bubbles come from?
[15] Figure 1 shows a red color for Cu-P25-ASP, whereas the other dispersions and solutions were not. Which solutions are mentioned. Do these contain copper?
[16] Table 1 shows Cu2+ decrease ratios. How these value are calculated? From which eperiment was copper concentration obtained. The results show that Cu2+ photoreducton is weak. Most of the Cu2+ is removed via Adsorption onto silica particles as ASP has 40% removal and in combination with P25 only 8% more. But why is only P25 so low?
Author Response
Dr. Francesco Baino
Special Issue Editor, “Porous Ceramics, Glasses and Composites” in Materials
Dear Dr. Baino
Thank you very much for your kind treatment of our manuscript titled “Photoreduction of copper ions using silica–surfactant hybrid and titanium (IV) oxide under sulfuric acid conditions” by by myself, R. Kato, K. Hasegawa, T. Gotoh, K. Katsumata, and A. Yasumori in Special Issue “Porous Ceramics, Glasses and Composites” in Materials. We would like to thank the reviewers for valuable comments and suggestions. We would like to reply to the inquiries as follows. Please note that changes based on the reviewer’s comments are marked in red in this reply as well as in the revised manuscript.
Reviewer: 1
The authors like to combine reduction of heavy metal ions and oxidation of surfactants both photocatalytically with TiO2 as the photocatalyst. This general approach is ok, but the whole story is not well presented. There are several lacks and I cannot recommend this manuscript for publication. This manuscript has not been prepared with the care and detail expected. Below are some comments that might help to improve the quality of the manuscript for future submission elsewhere.
Introduction:
[1] The reduction of heavy metals is clear, but why surfactants extracted from silica–surfactant hybrids as organic material and not surfactants in general? If only the surfactant should be degraded that comes from the synthesis (liquid phase) of porous silica, why where silica particles are used in the study?
Reply [1]:
We thank the reviewer 1 for pointing out this issue. We apologize the lack of motivation for using silica-surfactant hybrids. We have two motivations as follows. Calcination using electric furnace or acid extraction are necessary for eliminating surfactants from silica-surfactant hybrids to convert them into mesoporous silicas. Acid extraction of silica-surfactant hybrid is essential to avoid the use of electric furnace that emit CO2 gases, while acid extraction releases surfactants to waste water. Photocatalytic reaction is useful for purifying the waste water. In addition, mesoporous silicas with relatively high specific surface areas can be useful for adsorbent of heavy metal ions (Materials, 2012, 5, 2874.). The combinations of these two motivations matches the purpose of this study. Therefore, based on the reviewer 1 comment, we modified introduction to easily understand our motivation for readers.
We thus added the texts in the revised manuscript as follows.
・Introduction, 1st paragraph.
The discovery of mesoporous silicas prepared by a supramolecular templating approach in the 1990s1,2 prompted studies of their fundamental properties and practical applications, leading to the generation of mesoporous silicas with varied mesostructures that originated from the mesostructures of silica–surfactant hybrids that form via self-assembly of surfactants such as alkyltrimethylammonium salts with concurrent silica precipitation.1-7 Porous structures thus form upon removal of the surfactants via calcination using an electric furnace or via acid extraction under mild conditions.1-7 The former process can emit carbon dioxide gas, whereas the latter process proceeds under mild conditions but generates a waste solvent that contains surfactants. Therefore, we have focused on adding this surfactant-containing waste to wastewater to promote the removal of pollutants for environmental purification applications. Notably, mesoporous silicas that release surfactants can be used as adsorbents to purify waste water.6,7
[2] The target should be formulated more clear. "We like to reduce heavy metal ions and oxidize surfactants simultaneously by photocatalysis with TiO2 as the photocatalyst."
Reply [2]:
We thank the reviewer 1 for pointing out this issue. We thus added the text in the revised manuscript as follows.
・Introduction, 2nd paragraph.
Here, we report the photoreduction of Cu2+ ions using silica–surfactant hybrids and titanium (IV) oxide (TiO2) under sulfuric acid conditions (Scheme 1). Specifically, photoreduction of heavy-metal ions was conducted with the concurrent oxidation of surfactants by combining TiO2 with silica–surfactant hybrids. TiO2 is an extensively studied photocatalyst8-10 that promotes the photo-oxidation of organic compounds and the photoreduction of heavy-metal ions to metals. 8-11 The latter process can be used to produce visible-light-responsive photocatalysts and antibacterial materials, 8-11 and the generated heavy metals can be photo-oxidized by TiO2.12 Notably, heavy-metal ions are present in sulfuric-acid-containing wastewater from mines. 13, 14 Sulfate ions can be adsorbed onto the surface of TiO2 to decrease its photocatalytic activity.15 Therefore, surfactants extracted from silica–surfactant hybrids by sulfuric acid are promising organic compounds that can be degraded by photo-oxidation by TiO2, promoting the photoreduction of heavy-metal ions by TiO2 in the presence of sulfuric acid. Thus, the photocatalytic activity of TiO2 has the potential to both degrade surfactants present as additional waste and remove heavy-metal ions as metal precipitates.
Materials:
[3] Most information are missing. Quality and origin of used chemicals.
Reply [3]:
We thank the reviewer 1 for pointing out this issue. We apologize the lack of material information. We thus added information of chemicals in the revised manuscript as follows.
・Material and Methods, 1st paragraph in the revised manuscript.
Copper(II) sulfate (CuSO4) pentahydrate and methanol were obtained from Wako Pure Chemical. Tetraethoxysilane (TEOS) and hexadecyltrimethylammonium chloride (C16TAC) were obtained from TCI. A 28 wt% ammonia solution was obtained from Kanto Chemical. All the chemicals were reagent grade and used without further purification.
[4] Why was the solution during irradiation not continuously stirred? Particles might settle and will not be irradiated between.
Reply [4]:
We thank the reviewer 1 for pointing out this issue. When we conducted the present reaction under continuous stirring, the resultant solid did not appear red coloration. This is because UV right unlikely reached the settle of reaction vessel or a certain irradiation time for silica-surfactant hybrids and P25 at the top of the dispersion was necessary for proceeding the present reaction. Meanwhile, fortunately, particles were not immediately settled during the present reaction. Thus, we stirred the dispersion for a moment every hour.
We therefore changed and added the texts in the revised manuscript as follows.
・Material and Methods, 2nd paragraph (this part is 1st paragraph in the original manuscript).
In the present study, we conducted photoreduction of Cu2+ ions because their concentration is easily measured and estimated by UV–vis spectrophotometric analysis of the well-known blue-colored complex [Cu(NH3)4]2+.16 To easily determine the particle morphology, in accordance with a previous study, 17 we prepared spherical silica–surfactant hybrid particles via homogeneous precipitation of both silica and C16TAC in a methanolic solution containing ammonia. After TEOS was added to the methanol/water mixture containing ammonia (the TEOS:C16TAC:H2O:methanol:ammonia molar ratio was 1:0.4:774:1501:72 in the initial dispersion), the solution was shaken for 3 s and aged at room temperature for 20 h. After the reaction, the resultant solid was centrifuged, washed with methanol, and then dried for 80 °C for 1 day. A field-emission scanning electron microscopy (FE-SEM) image of the product is shown in Figure 1. For comparison, the as-synthesized spherical particles (ASP) were calcined at 550 °C for 20 h, similar to the procedure used in a previous study17 to prepare calcined spherical particles (CSP) as a nanoporous silica. The TiO2 was a standard photocatalyst (P25, Degussa). After both P25 (20 mg) and ASP (80 mg) were dispersed in a 10 mmol/L CuSO4 solution (5 mL) adjusted to pH 4 by the addition of sulfuric acid, the dispersion was irradiated with a He–Xe lamp for 4 h though a quartz plate. The irradiation area was the same as the width of the reaction vessel, and the distance between the top of the dispersion (whose height from the bottom of the reaction vessel was 1 cm) and the lamp was 0.5 cm. During irradiation, the dispersion was briefly stirred for every hour. The dispersion was also continuously stirred for comparison purposes. After the reaction, the resultant solid was centrifuged at 5000 rpm for 10 min and then washed with distilled water, to obtain the product denoted herein as Cu–P25–ASP. The resultant supernatant showed an increase in pH from 4 to 5 and contained bubbles on its surface, a common feature of surfactant aqueous solutions. The obtained solid was dried under a reduced pressure for 1 day. For comparison, these procedures were also applied to separate starting solutions containing P25 (20 mg), ASP (80 mg), and C16TAC (20 mg) individually.
・Results and Discussion, first part.
After irradiation, the P25–ASP mixed dispersion was dark-red (Figure 2), whereas the dispersions of P25 or ASP and the solution of C16TAC containing Cu2+ were not. Notably, when the P25-ASP mixed dispersion was continuously stirred, the red coloration was not observed. Because the powders used in the present study did not immediately settle in the reaction vessel, UV light likely did not reach the vessel bottom and a certain UV irradiation time was necessary to change the dispersion color. The X-ray diffraction (XRD) pattern for the Cu–P25–ASP shows reflections attributable to P25 and ASP as well as Cu metal; no reflections attributable to copper oxides are observed (Figure 3). Transmission electron microscopy (TEM) and scanning TEM (STEM) images of the Cu–P25–ASP show spherical particles (ASP) and ~30 nm particles of P25. The latter particles exhibit relatively dark and bright regions (Figure 4a and b). The light regions contain high concentrations of Cu, as evident in the corresponding energy-dispersive X-ray (EDX) mapping image (Figure 4d). In addition, the spherical particles and the ~30 nm particles of P25 contain Si and Ti, as revealed in the EDX mapping images, respectively (Figure 4e and f). The X-ray photoelectron spectrum of the Cu–P25–ASP (Figure 5) shows peaks at binding energies of 953.5, 943.0, 933.0, and 932.5 eV, which are attributed to Cu 2p1/2, Cu2+ satellite, Cu2+ 2p3/2, and Cu 2p3/2, respectively.18,19 The ratios of the decrease in Cu2+ concentration relative to those in the starting solutions are reported in Table 1. The dispersion that contained both P25 and ASP resulted in the largest decrease, whereas the Cu2+ concentration hardly decreased for the P25 dispersion or the C16TAC solution. By contrast, the ASP dispersion resulted in a 40% decrease in Cu2+ concentration.
[5] The photocatalytic experiment must be explained in much more detail. The results depend strongly on the chosen conditions. Distance lamp-reactor, top- and side-irradiation, ...
Reply [5]:
We thank the reviewer 1 for pointing out this issue. We apologize the lack of detailed conditions of photocatalytic experiment. This point was described in the reply to the reviewer 1’s comment [4] (see page 3 and 4 in this response letter).
[6] How was the concentration of cupper and surfactant measured? It is mentioned to complex Cu2+ with ammonia. With ammonia gas? Or ammonia solution? Was it measured with UV/vis spectrometry? Which machine, ...?
Reply [6]:
We thank the reviewer 1 for pointing out this issue. We apologize the technical word error and the lack of description. We used the famous complex compounds [Cu(NH3)4]2+ showing blue coloration (J. Phys. Conf. Ser., 2020, 1481, 012040.) and estimated the Cu2+ concentration by adsorption spectra with Lamber-Beer’s Low. Thus, we think that the abovementioned techniques are enough to estimate the Cu2+ concentration.
We thus changed the texts in the revised manuscript. This point was described in the reply to the reviewer 1’s comment [4] (see the last part of page 3 in this response letter).
[7] Why not ICP for Cu2+?
Reply [7]:
We thank the reviewer 1 for pointing out this issue. This point was reflected in the reply to the reviewer 1’s comment [6].
[8] How was the concentration of the surfactant measured?
Reply [8]:
We thank the reviewer 1 for pointing out this issue. We apologized the lack of characterization of silica-surfactant hybrid. The surfactant content in silica-surfactant hybrid was estimated using CHN analysis.
We thus added the description of CHN analysis in the revised manuscript as follows.
・Page 9, 1st paragraph and page 10 1st paragraph in the revised manuscript (this part was Page 7, 1st paragraph and page 8 1st paragraph in the original manuscript).
The intensities of the C–H stretching bands20 at 2922 and 2872 cm−1 and the C–N stretching band20 at 1460 cm−1 in the Fourier transform infrared (IR) spectrum of the solid mixture of P25 (20 mg) and ASP (80 mg) are substantially lower than those in the spectrum of the Cu–P25–ASP (Figure 7). Based on the CHN analyses, the C and N contents (2.9 and 0.13 mass%, respectively) in the Cu–P25–ASP are lower than those in the mixed solid before irradiation (18 and 0.95 mass%, respectively). Figure 6 shows N2 adsorption/desorption isotherms for the ASP, P25, Cu–P25–ASP, and CSP, recorded at −196°C. Based on these isotherms, the specific surface areas of the ASP, P25, Cu–P25–ASP, and CSP were calculated to be 36, 45, 591, and 806 m2/g using the Brunauer–Emmett–Teller (BET) method. 21 These BET surfaces areas, the Barrett–Joyner–Halenda (BJH)22 pore sizes, and the pore volumes estimated using the BJH method are listed in Table 2 for Cu–P25–ASP and CSP.
Results:
[9] Figure 3, JCPDS for P25 (one for the anatase and one for the rutile phase)
Reply [9]:
We thank the reviewer 1 for pointing out this issue. Since we think that JCPDS for P25 is not present, we marked reflections due to anatase and rutile, respectively, in Figure 3 in the revised manuscript.
We thus changed Figure 3 in the revised manuscript as follows.
・Figure 3
Original: Revised:
Figure 3. XRD patterns for (a) ASP, (b) P25, and (c) Cu–P25–ASP. |
Figure 3. XRD patterns for (a) ASP, (b) P25, and (c) Cu–P25–ASP. Filled circles and triangles denote anatase and rutile reflections, respectively. |
[10] Figure 4, When Cu-P25-ASP is a composite (adsorbed Cu2+), then the EDX mapping should be also done for Si and Ti.
Reply [10]:
We thank the reviewer 1 for pointing out this issue. As the reviewer 1 mentioned, EDX mappings of Si and Ti are necessary for readers who are not familiar with silica-surfactant hybrids and titania.
We thus changed Figure 4 in the revised manuscript as follows. The change of the text in the revised manuscript was described in the reply to the reviewer 1’s comment [4] (see the last part of page 4 in this response letter).
・Figure 4
Figure 4. (a) TEM and (b) STEM (b) images of Cu–P25–ASP. The yellow square in (b) is enlarged in (c). The EDX mapping image of (c) is shown in (d). |
Original:
Figure 4. (a) TEM and (b) STEM (b) images of Cu–P25–ASP. The region indicated by the yellow square in (b) is enlarged in (c). EDX mapping image for Cu, Si, and Ti in figure (c) are shown in figures (d), (e), and (f), respectively. |
Revised:
[11] There is no Figure 6 as refereed in the text. It is named Figure S2, obviously taken from any Supporting Information.
Reply [11]:
We thank the reviewer 1 for pointing out this issue. We apologize the description error. Figure 6 is true. This point was reflected in the revised manuscript.
[12] All data from N2 adsorption (BET, pore size and volume) should be presented in a Table.
Reply [12]:
We thank the reviewer 1 for pointing out this issue. Based on the reviewer 1’s comment, pore size and volume that are estimated using BJH method (J. Am. Chem. Soc., 1951, 73, 373.) and BET surface areas are listed in Table 2. This point was reflected to the revised manuscript, and the modified text was described in the reply to the reviewer 1’s comment [8] (see the middle part of page 6 in this response letter).
[13] In the text, elementary analysis and IR spectra are mentioned that show the removal of the surfactant. However, this manuscript does not contain the data from IR and elementary analysis.
Reply [13]:
We thank the reviewer 1 for pointing out this issue. Based on the reviewer 1’s comment, we added the IR spectra as Figure 7 as follows in the revised manuscript. As for the elemental analysis, this point was described in the reply to the reviewer 1’s comment [8] (see page 8 in this response letter). The texts added to the revised manuscript was described in the reply to the reviewer 1’s comment [8] (see page 8 in this response letter).
Figure 7. IR spectra of ASP– P25 mixed solid (bottom) and Cu–P25–ASP (top). |
・Figure 7
[14] After the dispersion containing P25 and ASPs was irradiated for 4 h, slight bubbling was observed. And where does the bubbles come from?
Reply [14]:
We thank the reviewer 1 for pointing out this issue. The generation of bubbles is a common feature of surfactant aqueous solution, because the surfactants are extracted from silica-surfactant hybrids under acidic condition in this study. This point was reflected in the reply to the reviewer 1’s comment [4] (see page 4 in this response letter).
[15] Figure 1 shows a red color for Cu-P25-ASP, whereas the other dispersions and solutions were not. Which solutions are mentioned. Do these contain copper?
Reply [15]:
We thank the reviewer 1 for pointing out this issue. We apologize the lack of description. We mentioned the dispersions of P25 or ASP and the solution of C16TAC. In addition, these dispersions and solution contained Cu2+. This point was reflected in the reply to the reviewer 1’s comment [4] (see page 4 in this response letter).
[16] Table 1 shows Cu2+ decrease ratios. How these value are calculated? From which experiment was copper concentration obtained. The results show that Cu2+ photoreducton is weak. Most of the Cu2+ is removed via Adsorption onto silica particles as ASP has 40% removal and in combination with P25 only 8% more. But why is only P25 so low?
Reply [16]:
We thank the reviewer 1 for pointing out this issue. The former point was described in the reply to the reviewer 1’s comment [6] (see page 6 in this response letter). As for the low adsorption capability of P25, the sulfate ion generally suppresses adsorption/photocatalytic reactions of titania (J. Chem. Technol. Biotechnol., 2001, 77, 102: J. Colloid Interface Sci., 1973, 44, 430.). Since all the photocatalyst experiment in this study were conducted under sulfate acid acidic condition, the Cu2+ adsorption capability of P25 is low.
We therefore added the texts in the revised manuscript as follows.
・Results and Discussion, last paragraph.
After the dispersion containing P25 and ASP was irradiated for 4 h, slight bubbling was observed on the surface of the dispersed particles, suggesting the presence of undegraded C16TMA+. In addition, C16TMA+ and/or C16TAC was incompletely extracted from the ASP according to the residual alkyl chains in Cu–P25–ASP, as revealed by the IR spectra and CHN analyses; this interpretation is further supported by the lower porosity of Cu–P25–ASP compared with that of CSP, as determined from the N2 adsorption/desorption isotherms (Figure 6). Notably, silica–surfactant hybrid spheres have been prepared using alkyltrimethylammonium salts with shorter side-chain lengths.4 In addition, the synthesis of silica–surfactant hybrids has been used to coat silica–surfactant layers onto various inorganic surfaces to form inorganic solid/silica–surfactant core–shell particles.5,19 Increasing the amounts of TiO2 and TiO2-based compounds26, 27 and/or other materials28 added to the present reaction is also feasible. Meanwhile, because the Cu2+ adsorption capability of P25 is low under sulfuric acid conditions (Table 1), the cation adsorption ability of TiO2 surfaces in the presence of protons, where acid sites on TiO2 surfaces could be strongly related to their adsorption ability, warrants further investigation.29-32 However, the present product containing Cu metal might find applications where visible-light-responsive photocatalysts and antibacterial materials7-14 have been used. We plan to conduct such studies in the future.
Reviewer 2
In this manuscript, photoreduction of Cu2+ ions to Cu metal by TiO2 was facilitated under sulfuric acid conditions, where the surfactants extracted from silica–surfactant hybrids by protons in the acidic condition were successfully photo-oxidized by TiO2. The formation of metal Cu can be well confirmed by the results of XRD, XPS, etc. Such a material system designed for the recovery of heavy metal ions in aqueous solutions may be appealing to the wide readership of this journal, whereas a major revision is still needed for final publication. The concerns are listed as follows:
- Detailed experimental conditions for the preparation of spherical silica–surfactant hybrid particles can be added.
Reply 1:
We thank the reviewer 2 for pointing out this issue. We apologize the lack of the experimental condition.
We thus added the texts in the revised manuscript as follows.
・Material and Methods, 2nd paragraph (this part is 1st paragraph in the original manuscript).
In the present study, we conducted photoreduction of Cu2+ ions because their concentration is easily measured and estimated by UV–vis spectrophotometric analysis of the well-known blue-colored complex [Cu(NH3)4]2+.16 To easily determine the particle morphology, in accordance with a previous study, 17 we prepared spherical silica–surfactant hybrid particles via homogeneous precipitation of both silica and C16TAC in a methanolic solution containing ammonia. After TEOS was added to the methanol/water mixture containing ammonia (the TEOS:C16TAC:H2O:methanol:ammonia molar ratio was 1:0.4:774:1501:72 in the initial dispersion), the solution was shaken for 3 s and aged at room temperature for 20 h. After the reaction, the resultant solid was centrifuged, washed with methanol, and then dried for 80 °C for 1 day. A field-emission scanning electron microscopy (FE-SEM) image of the product is shown in Figure 1. For comparison, the as-synthesized spherical particles (ASP) were calcined at 550 °C for 20 h, similar to the procedure used in a previous study17 to prepare calcined spherical particles (CSP) as a nanoporous silica. The TiO2 was a standard photocatalyst (P25, Degussa). After both P25 (20 mg) and ASP (80 mg) were dispersed in a 10 mmol/L CuSO4 solution (5 mL) adjusted to pH 4 by the addition of sulfuric acid, the dispersion was irradiated with a He–Xe lamp for 4 h though a quartz plate. The irradiation area was the same as the width of the reaction vessel, and the distance between the top of the dispersion (whose height from the bottom of the reaction vessel was 1 cm) and the lamp was 0.5 cm. During irradiation, the dispersion was briefly stirred for every hour. The dispersion was also continuously stirred for comparison purposes. After the reaction, the resultant solid was centrifuged at 5000 rpm for 10 min and then washed with distilled water, to obtain the product denoted herein as Cu–P25–ASP. The resultant supernatant showed an increase in pH from 4 to 5 and contained bubbles on its surface, a common feature of surfactant aqueous solutions. The obtained solid was dried under a reduced pressure for 1 day. For comparison, these procedures were also applied to separate starting solutions containing P25 (20 mg), ASP (80 mg), and C16TAC (20 mg) individually.
- Actually, “~30 nm particles of P25” cannot be found in the TEM images. And further please add the EDS mapping image of Ti element.
Reply 2:
We thank the reviewer 2 for pointing out this issue. This point was described in the reply to the reviewer 1’s comment [10] (see pages 7 and 8 in this response letter).
- The quality of the figures such as XRD, XPS can be further improved. Please refer to https://doi.org/10.1016/j.cej.2020.125347
Reply 3:
We thank the reviewer 2 for pointing out this issue. As for the XRD patterns, this point was described in the reply to the reviewer 1’s comment [9] (see pages 6 and 7 in this response letter). Concerning the XPS spectrum, the empty space at the upper part of the spectrum was removed. In addition, we cited the article presented by the reviewer 2 in the revised manuscript. These points were reflected in the revised manuscript.
- The performance for Cu2+ reduction can be compared with the previous studies focused on TiO2-based photocatalysts.
Reply 4:
We thank the reviewer 2 for pointing out this issue. As the reviewer 2 mentioned, we wanted to compere our results to the previous studies. However, there are no detailed and basic studies to conduct photocatalyst reactions under sulfuric acid acidic conditions. In addition, such the studies will be strongly related to the surface chemistry of titania (J. Phys. Chem. C, 2010, 114, 21531; Phys. Chem. Chem. Phys., 2012, 14, 9468; Chem. Mater., 2013, 25, 385.). We think that these will be another study.
We therefore added the texts in the revised manuscript as follows.
・Results and Discussion, last paragraph.
After the dispersion containing P25 and ASP was irradiated for 4 h, slight bubbling was observed on the surface of the dispersed particles, suggesting the presence of undegraded C16TMA+. In addition, C16TMA+ and/or C16TAC was incompletely extracted from the ASP according to the residual alkyl chains in Cu–P25–ASP, as revealed by the IR spectra and CHN analyses; this interpretation is further supported by the lower porosity of Cu–P25–ASP compared with that of CSP, as determined from the N2 adsorption/desorption isotherms (Figure 6). Notably, silica–surfactant hybrid spheres have been prepared using alkyltrimethylammonium salts with shorter side-chain lengths.4 In addition, the synthesis of silica–surfactant hybrids has been used to coat silica–surfactant layers onto various inorganic surfaces to form inorganic solid/silica–surfactant core–shell particles.5,19 Increasing the amounts of TiO2 and TiO2-based compounds26, 27 and/or other materials28 added to the present reaction is also feasible. Meanwhile, because the Cu2+ adsorption capability of P25 is low under sulfuric acid conditions (Table 1), the cation adsorption ability of TiO2 surfaces in the presence of protons, where acid sites on TiO2 surfaces could be strongly related to their adsorption ability, warrants further investigation.29-32 However, the present product containing Cu metal might find applications where visible-light-responsive photocatalysts and antibacterial materials7-14 have been used. We plan to conduct such studies in the future.
- The effect of pH value on the reduction of Cu2+ ions can be further investigated.
Reply 5:
We thank the reviewer 2 for pointing out this issue. We also think that the effect of pH value is one of the interests of this study, while the surface chemistry of titania should be concurrently investigated. This point was also described in the reply to the reviewer 2’s comment 4 (see this page).
- After the photoreduction reaction, the pH of the aqueous solution was increased. Please explain it.
Reply 6:
We thank the reviewer 2 for pointing out this issue. An increase in pH of the aqueous solution is likely due to the exchange reaction of C16TMA+ and protons in the sulfuric acid acidic solution. The reaction is very famous.
We thus added the text in the revised manuscript as follows.
・Page 11, 2nd paragraph in the revised manuscript (this part was page 8, 3rd paragraph and page 9 1st paragraph in the original manuscript).
Alkylammonium ions are well known to be degraded by •OH and •O2 radicals, which can be generated by reaction of H2O and O2 molecules with holes photogenerated in the valence band of TiO2.24 In the present study, the IR spectra, N2 adsorption/desorption isotherms (Figure 6), and elemental analyses indicate a decrease in the concentration of hexadecyltrimethylammonium ions (C16TMA+) in the ASP after the reaction. Because a decrease in the concentration of Cu2+ was observed when ASP alone were used for the present reaction (Table 1) and because the cation-exchange reactions of silanol groups on silicas are well known, 25 exchange reactions of C16TMA+ with protons and/or Cu2+ ions in the CuSO4 acidic solution are highly likely, as indicated by an increase in the pH of the dispersion after the reaction (see Material and Methods). The presence of Cu2+ detected by XPS (Figure 5) thus results from Cu2+ ions adsorbed onto the ASP, which are not accessible on P25 surfaces (Figure 4) or on the white regions observed after irradiation (Figure 2); the latter situation can be improved by modifying the experimental conditions. The dispersion never became dark-red as shown in Figure 2 when the present reaction was conducted with P25 alone; it remained white. In addition, the Cu2+ decrease ratio for the Cu–P25–ASP is higher than those for the P25 and ASP. Notably, the solution after irradiation was still under sulfuric acid conditions (refer to the experimental conditions). Therefore, photo-oxidation of C16TMA+ extracted from ASP is evident in the present reaction, and this photo-oxidation successfully promoted the photoreduction of Cu2+ under sulfuric acid conditions.
- It is suggested to add a schematic diagram including the structural characteristics of the designed materials and the photoreduction process of Cu ions over the TiO2 photocatalyst.
Reply 7:
We thank the reviewer 2 for pointing out this issue. Based on the reviewer 2’ comment, we added Scheme 1 as follows and as an overview of the present reaction at the first part of 2nd paragraph in Introduction in the revised manuscript.
Scheme 1. An overview of the present study. |
In addition to the change for the reviewer’s comments, we have made the following changes, some of which are directly marked in the revised manuscript in blue.
- Typological have been corrected.
- Unification of terms has done.
- References have been added. Reference numbers have been adequately adjusted.
Improvement of English has been attempted, because the reviewers pointed out English quality. Native check letter is shown below.
We marked all changes in red in the revised manuscript. We would like to thank the reviewers again for the comments and valuable suggestions. Some of these comments provide us new and deep insights of our results. We hope that the revised manuscript is acceptable for publication in Special Issue “Porous Ceramics, Glasses and Composites” in Materials
Yours sincerely,
Shingo Machida, Ph. D.
Assistant Professor
Department of Material Science and Technology
Tokyo University of Science
6-3-1, Katsushika-ku, Tokyo 125-8585, Japan
Phone: +81-3-5876-1717
E-mail: shingo.machida@rs.tus.ac.jp

Reviewer 2 Report
Manuscript-ID:materials-1781653
Title:Photoreduction of copper ions using silica–surfactant hybrid and titanium (IV) oxide under sulfuric acid conditions
In this manuscript, photoreduction of Cu2+ ions to Cu metal by TiO2 was facilitated under sulfuric acid conditions, where the surfactants extracted from silica–surfactant hybrids by protons in the acidic condition were successfully photo-oxidized by TiO2. The formation of metal Cu can be well confirmed by the results of XRD, XPS, etc. Such a material system designed for the recovery of heavy metal ions in aqueous solutions may be appealing to the wide readership of this journal, whereas a major revision is still needed for final publication. The concerns are listed as follows:
1. Detailed experimental conditions for the preparation of spherical silica–surfactant hybrid particles can be added.
2. Actually, “~30 nm particles of P25” can not be found in the TEM images. And further please add the EDS mapping image of Ti element.
3. The quality of the figures such as XRD, XPS can be further improved. Please refer to https://doi.org/10.1016/j.cej.2020.125347
4. The performance for Cu2+ reduction can be compared with the previous studies focused on TiO2-based photocatalysts.
5. The effect of pH value on the reduction of Cu2+ ions can be further investigated.
6. After the photoreduction reaction, the pH of the aqueous solution was increased. Please explain it.
7. It is suggested to add a schematic diagram including the structural characteristics of the designed materials and the photoreduction process of Cu ions over the TiO2 photocatalyst.
Author Response

(The authors gave the same response as above.)

Round 2
Reviewer 2 Report
The previous comments have been well addressed and the revised manuscript can be accepted for publication.